# Modeling the Impact of Ecological Restoration on Waterbird Diversity and Habitat Quality in Myanmar’s Moe Yun Gyi Wetland

**DOI:** 10.3390/biology14050519

**Published:** 2025-05-08

**Authors:** Phyoe Marnn, Haider Ali, Haibo Jiang, Yang Liu, Ziqi Li, Sarfraz Ahmed, Tao Yang, Ziwei Li, Chunguang He

**Affiliations:** 1Key Laboratory of Wetland Ecology and Vegetation Restoration, Ministry of Ecology and Environment, Northeast Normal University, Changchun 130117, China; feie337@nenu.edu.cn (P.M.); haid555@nenu.edu.cn (H.A.); liuyuan100@nenu.edu.cn (Y.L.); hamd@nenu.edu.cn (S.A.); yangt843@nenu.edu.cn (T.Y.); lizw830@nenu.edu.cn (Z.L.); 2Key Laboratory for Vegetation Ecology, Ministry of Education, Northeast Normal University, Changchun 130117, China; 3College of Modern Agriculture and biotechnology, Ankang University, Ankang 725000, China; 13720789522@163.com; 4Key Laboratory of Remote Sensing, Northeast Institute of Geography and Agroecology, Chinese Academy of Sciences, Changchun 130102, China

**Keywords:** migrating, populations, efforts, rehabilitation, hydrology

## Abstract

The study examined the impact of work to restore one wetland in Myanmar, known as the Moe Yun Gyi site, on bird life from 2014 to 2023. This wetland is significant due to it sitting along a significant bird migration route. The number of waterbirds had multiplied by sevenfold over a decade and there were more birds than ever recorded, including some that are considered to be at risk. The enhancements involved the diversion of 25 million cubic meters of water into the wetland and the restoration of healthy marsh areas. The alterations made for birds were even more conducive to the animals and correspondingly led to a peak in bird diversity in 2020. Bird counts and mapping tools were used to follow the changes. Flooding in 2023 resulted in a decrease in bird numbers, but overall, the project is evidence that restoring wetlands can have a big impact on birds and other wildlife. And if restoration efforts proceed, predictions say that the place should be that much more habitable for birds by 2040. The study illustrates the importance of continued care for wetlands in order to conserve nature and support migrating bird flights.

## 1. Introduction

Wetland restoration is a global endeavor, with significant efforts being made in America, Europe, and Asia. Wetlands are one of the most valuable ecosystems on the planet, providing key habitats for various species, supporting climate regulation, and playing an important role in biodiversity conservation [1,2]. But, for more than 45 years, almost 35% of natural wetlands have been lost due to human activities, seriously threatening the lives of migratory and resident birds [3,4]. Restoration aims to alleviate wetlands’ loss and deterioration by facilitating their return to a more authentic condition [5]. The Moe Yun Gyi wetland serves as a crucial environment for both resident and migratory birds. It is designated an Important Bird and Biodiversity Area (IBA) because it is home to globally endangered bird species (the Great Knot (*Calidris tenuirostris*), Black-faced Spoonbill (*Platalea minor*), Great Crested Grebe (*Podiceps cristatus*), White-naped Crane (*Grus vipio*), Black-tailed Godwit (*Limosa limosa*), Spot-billed Pelican (*Pelecanus philippensis*), and Philippine Duck (*Anas luzonica*)). This designation highlights the importance of wetland conservation. Moe Yun Gyi Wetland, in particular, is recognized as an ecological hotspot, supporting a diverse range of bird species. Additionally, the wetlands attract large groups of migratory bird species [6]. Moreover, aquatic birds, sometimes known as waterbirds, rely entirely on wetlands for various behaviors, such as searching for food, resting, and shedding feathers [7].

Birds are widely considered reliable indicators of overall ecosystem health and condition due to their high sensitivity to environmental changes [8,9]. Understanding bird diversity patterns is crucial for informing and supporting conservation management. Birds fulfil various functional roles, making the study of their diversity patterns essential [8]. Waterbirds exhibit a higher level of sensitivity to climate conditions in comparison to other species of birds [8]. Because of the worldwide threat that expanded agricultural land use poses to avifauna, integrating agroecosystems as bird habitats is critical for long-term conservation planning. Most waterbird species are gregarious species that rely on wetlands for their existence, and they play a vital role in wetland ecosystems [9,10]. The community composition and variety of wetlands serve as a direct reflection of changes in the wetland ecosystems. They also function as an objective biological indicator to monitor changes in the wetland environment [10]. Waterbirds exhibit distinct seasonal migration patterns [9]. Waterbird migration effectiveness depends on a sequence of wetland stopover locations available along the migration route. The species richness and abundance of wintering waterbirds, especially migratory species, are influenced by environmental conditions that may not impact other organisms [9,11,12]. Terrestrial ecosystems can be effectively measured using several indicators, including biomass, NDVI, and potential evapotranspiration [13,14,15]. These indicators serve as a proxy for energy flux in terrestrial systems and are readily accessible for analysis. Waterbirds play a crucial role in aquatic systems and significantly impact the ecological function of wetlands. Alterations in waterbird populations can lead to corresponding modifications in other organisms within wetland ecosystems, resulting in the transmission of disturbance effects across aquatic food chains and influencing the cycling of nutrients [15]. The Moe Yun Gyi Wetland of Myanmar, which is located along the East Asian–Australasian Flyway, is an important stopover site for migratory waterbirds. This wetland, whilst essential, is also under threat from agricultural expansion, pollution and climate change, and deserves urgent conservation effort. The Moe Yun Gyi Wetland is designated as an Important Bird Area (IBA) because of its substantial bird population. The wetland is threatened significantly by agricultural expansion, pollution from regional and transboundary sources and habitat degradation [6]. This designation protects the conservation and biodiversity of the wetland, which is increasingly threatened by encroachment for farming, pesticide runoff and climate change [16,17]. It is crucial to protect this profuse habitat from persistent threats that require conservation measures [18,19]. Conservation action needs to be taken to safeguard this important habitat from continual threats. Before restoration measures were implemented, problems in the Moe Yun Gyi wetland area were caused by unreasonable exploitation of wetland, grassland, and water resources such as reduced grassland areas and shrinking wetland regions. Restored wetlands have been shown to increase waterbird abundance, enhance species diversity, improve breeding success, and provide critical stopover or wintering habitats by recreating suitable hydrological conditions, increasing food availability, and restoring native vegetation [1]. In 2017, restoration efforts began with the mission of reconnecting fragmented habitats in the Moe Yun Gyi wetland. These actions reconnected some isolated lakes, Pakaing reservoirs, and seasonal and dry marshes. As a result, the area of water bodies and wetlands in the reserve increased, providing suitable habitats for the waterbird community. During the initial phase of restoration from 2017 to 2019, the annual water diversion in the region was approximately 15 million m^3^. This diversion amount was reduced to around 12 million m^3^ from 2020 to 2022. By the end of the restoration measures, water diversion levels remained consistent with those at the beginning of the restoration efforts.

Our study shows that restoration has significant effects focus on waterbird populations. They highlight the need for efforts to manage water. We hypothesized that ecological restoration in Moe Yun Gyi Wetland significantly enhanced the diversity and abundance of rare waterbird species, especially those sensitive to habitat conditions. This study aimed to investigate the impacts of restoration actions to plan in the period 2014–2023 on waterbird diversity, population trends, and habitat stability. On the basis of known and projected trends (2030, 2040), we formulated well-founded advice on sustainable wetland management and community-based management for biodiversity conservation. These results underscore the importance of water-level control and hydrology for the maintenance of favorable wetland conditions. It will be this study that will prove to be invaluable in determining the success of different conservation measures and that will inform active wetland management in the future. Additionally, the research partners ecological, social, and economic considerations to enhance the resilience and sustainability of habitat for migratory birds, thus promoting a science-based management and protecting strategic stopover sites across migratory flyways.

## 2. Materials and Methods

### 2.1. Study Area

Moe Yun Gyi Wetland is located northeast of Pyinbongyi village in Myanmar’s Bago Region (16°45′ N, 95°34′ E), approximately 25 km from Bago city (Figure 1). Spanning 103 km^2^, it is an important ecological zone that provides essential habitat to many resident and migratory waterbird species. The various habitats that abound there provide important feeding and breeding grounds for numerous species of waterbirds, including those that reside there throughout the year and those that migrate.

Sustainable management practices, including wetland rehabilitation, habitat construction, water regulation, and invasive species control, have been employed to restore natural habitats and enhance ecosystem services, supporting waterbird diversity. The Moe Yun Gyi Wetland is well known for its rich birdlife and serves as an important refuge for both endemic and migratory bird species. It has been undergoing conservation since 2017, which involved habitat restoration, wetland reconstruction, water level regulation, and prevention and control of invasive species. It is also a major birdwatching and ecotourism site in Myanmar. Before adaptation, significant ecological degradation occurred in the wetland due to overuse of natural resources, such as land shrinkage and grassland loss. The purpose of adaptation was to rehabilitate fragmented aquatic habitats such as isolated lakes, the Pakaing reservoir, and seasonal marshes, which collectively expanded suitable areas for waterbirds.

The first period (2017–2019) addressed emergency habitat remediation, including the clearing of water channels, replenishment of seasonal marshes, and the re-routing of around 15 million m^3^ of water annually to achieve hydrological balance. During the second phase (2020–2022), the focus was on ecological stabilization, including invasive species management, planting wetland vegetation, and adaptive management to maintain water levels for bird habitat. Later adjustments reduced the diversion capacity to 12 million m^3^/year based on annual hydrological requirements. The third stage (from 2023) focuses on building resilience and monitoring, incorporating long-term water regulation regimes and the tracking of biodiversity. These staged interventions aimed to progressively restore wetland structure, enhance habitat heterogeneity, and support the recovery of wildlife populations.

### 2.2. Habitat Landscape in Remote Sensing Interpretation

Landsat 5 TM and Landsat 8 OLI images with a path-row number of 120–029 were chosen for habitat types and surface water classification in the Moe Yun Gyi Wetland. These CT images were collected between 1 January 2014 and 1 January 2024. Imagery was favored during dry seasons (January–March) to minimize cloud-related distortion and to enhance classification accuracy. All the images were processed in a standard way, including atmospheric correction, cloud and shadow masking by the Fmask algorithm [9], and radiometric calibration.

The use of multitemporal imagery was pursued to enhance object distinction and land cover interpretation. The habitat was classified by means of a maximum likelihood supervised classification, and the accuracy of the classification was evaluated with ground-truth points and confusion matrices. The overall accuracy of classification was more than 85% and the Kappa coefficient was over 0.80.

Water level and diversion data for Moe Yun Gyi Wetland were obtained from internal reports provided by the Department of Irrigation and Water Utilization Management, Ministry of Agriculture, Livestock and Irrigation, Myanmar. These government-sourced data were acquired through official communication and are not available online.

### 2.3. Bird Survey Methods

A total of 20 observation sites were established in the study region based on the historical stopover sites of waterbirds, including wetland habitats, using the Point Count Method for bird monitoring (Figure 2). The waterbirds in the area were categorized into four distinct groups according to their habitat preferences: dabbling birds, diving birds, large waders, and small waders. The dabbling birds primarily consist of *Anas acuta* (*Linnaeus, 1758*), *Anas crecca* (*Linnaeus, 1758*), *Anas poecilorhyncha* (*Forster*, *1781*), *Anas querquedula* (*Linnaeus*, *1758*), *Nettapus coromandelianus* (*Gmelin*, *1789*), *Sarkidiornis melanotos* (*Pennant*, *1769*), and *Tadorna ferruginea* (*Pallas*, *1764*). The diving birds include *Anhinga melanogaster* (*Pennant*, *1769*), *Phalacrocorax carbo* (*Linnaeus*, *1758*), and *Tachybaptus ruficollis* (*Pallas*, *1764*). The group of large wading birds mainly consists of *Anastomus oscitans* (*Boddaert*, *1783*), *Anhinga melanogaster* (*Pennant*, *1769*), *Ardea cinerea* (*Linnaeus*, *1758*), *Ardea purpurea* (*Linnaeus*, *1766*), *Egretta egretta* (*Linnaeus*, *1766*), *Mesophoyx intermedia* (*Wagler*, *1829*), *Mycteria leucocephala* (*Latham*, *1790*), *Pelecanus philippensis* (*Gmelin*, *1789*), *Phalacrocorax niger* (*Vieillot*, *1817*), and *Threskiornis melanocephalus* (*Latham*, *1790*). The small waders are primarily composed of *Calidris temminckii* (*Leisler*, *1812*), *Charadrius dubius* (*Scopoli*, *1786*), *Charadrius veredus* (*Gould*, *1848*), *Gallinago gallinago* (*Linnaeus*, *1758*), *Limosa limosa* (*Linnaeus*, *1758*), *Tringa ochropus* (*Linnaeus*, *1758*), *Tringa glareola* (*Linnaeus*, *1758*), and *Tringa totanus* (*Linnaeus*, *1758*).

Waterbirds were monitored in the research area throughout the year, from January to December, from 2014 to 2024, based on their migration pattern.

Each stage of the survey included two observations conducted 3 to 4 days apart. Surveys were carried out systematically by scanning the sample areas using 8 × 32 binoculars and 20 × 60 spotting scopes, moving in a clockwise direction. Observations were made from fixed vantage points during peak bird activity, from 06:30 to 12:30 daily. Birds were primarily counted while they were stationary (on water or land); individuals in flight were also counted if their altitude was low enough to allow accurate identification, generally below 50 m. High-flying birds that could not be reliably identified were excluded from the dataset to maintain accuracy. Bird species and numbers were recorded, with point counts used for species in lower abundance, and flock estimates made in groups of 10, 20, or 50 for large congregations [9,10]. Sex and age were not distinguished, as many species lack obvious sexual dimorphism or were observed at distances that prevented accurate age or sex classification.

Species identification followed Birds of Myanmar and other authoritative guides, and conservation status assessments were based on data from the IUCN Red List, BirdLife International, and BANCA. To avoid double-counting, survey points were spaced at least 4 km apart, and a minimum of 20–25% of the wetland area was covered to ensure representative sampling. The duration at each observation site varied according to habitat size, bird density, and visibility [10,20]. Relative abundance, species diversity, density, and encounter rates were calculated using the point count method.

### 2.4. Data Analysis

Data Processing of Waterbird Community

The population size level is determined using the Berger–Parker dominance index, and the calculation formula is as follows:(1)$$I=\frac{Ni}{N}$$
where *Ni* is the number of individuals of species *i* and *N* is the total number of all species in the community. When I ≥ 0.1, species *i* is a dominant species, 0.01 ≤ I ≤ 0.1 indicates species *i* is a common species, and I < 0.01 indicates species *i* is an occasional species.

Using the Shannon-Wiener diversity index, Pielou evenness index, Simpson dominance index, and G–F index, a diversity analysis of the waterbird community in the region was conducted to understand the composition and structural changes of the bird community in the Moe Yun Gyi Wetland. The Shannon-Wiener index is used to reflect the species richness and evenness of a community. The G–F index is a species diversity parameter based on the number of species studied at the family and genus levels, reflecting the species diversity of a region over a long period. The G-index, F-index, and G–F index summarize waterbird species composition information.

Shannon-Wiener Refers to (H’)

The Shannon-Wiener diversity index reflects both the richness (number of species) and the evenness (distribution of individuals among species) of a community. It is commonly used to quantify biodiversity in ecological studies.H’ = −∑ Pi.LnPi(2)
where
H’ = Species DiversityPi = the proportional of individuals found in the ith speciesln = the natural logarithmH’ between 1.5 and 3.5 is found in natural communities.
Pielou Uniformity Index (E)

The Pielou Evenness Index measures the uniformity of species abundances, that is, how evenly individuals are distributed across the different species. A high value indicates a more balanced community structure.(3)$$E=\frac{H’}{Hmax}$$
Simpson Advantage Index (C)

The Simpson Dominance Index emphasizes the dominance of particular species in a community. It gives more weight to common or abundant species, thus indicating whether a few species dominate the community.(4)$$C=\sum_{i=1}^{S}{pi}^{2}$$

In the formula, *p* represents the proportion of individuals of species *i* to the total number of individuals of all species, *S* represents the total number of waterbirds in the study area, and Hmax = Ln*S*.

G-index (D_G_)

The G-index (Genus Diversity) evaluates the taxonomic diversity at the genus level, reflecting how species are distributed across different genera.(5)$$D_{G}\;=-\sum_{j=1}^{p}D_{Gj}=-\sum_{j=1}^{p}q_{j}\mathrm{Ln}q_{j}$$

In the formula, *q_j_* = *S_j_*/*S*, where *S_j_* represents the number of species in the *j* genus of waterbirds in the study area, *S* represents the total number of species in the waterbirds in the study area, and *p* represents the number of waterbird genera in the study area.

F-Index (D_F_)

The F-index (Family Diversity) evaluates the taxonomic diversity at the family level, measuring how evenly species are spread across families.(6)$$D_{F}\;=-\sum_{k=1}^{m}D_{FK}$$(7)$$D_{\mathrm{FK}}=-\sum_{j=1}^{p}p_{i}\mathrm{Ln}p_{i}$$

In the formula, *p_i_* = *S_ki_*/*S_k_*, where *S_k_* represents the number of species in the *k*-family of waterbirds, *S_ki_* represents the number of species in the *i*-genus of the *k*-family of waterbirds, and n represents the number of genera in the *k*-family of waterbirds in the study area. M represents the number of waterbird families.

Diversity Gain and Frequency Index, G–F Index (D_G–F_)

The G–F index integrates genus and family diversity to give a composite view of long-term taxonomic diversity and evolutionary distinctiveness of the waterbird community.(8)$$D_{G–F}\;=1-\frac{D_{G}}{D_{F}}$$

### 2.5. A Methodological Approach for 2030 and 2040 Projections

This methodology combined historical data analysis with machine learning and scenario-based modeling to predict habitat changes as well as waterbird diversity for the years 2030 and 2040. This method was utilized to permit thorough data-driven analysis of future ecological states under simulated alternative management strategies. By combining this with machine learning and scenario modeling, we could quantify anticipated land-cover changes and infer the associated impacts on waterbird diversity. An integrated approach is suitable for forecasting under non-linear interactions between environmental variables and species distributions, providing useful information for long-term conservation planning and adaptive wetland management.

Historical land cover and waterbird diversity data from 2010 to 2020 were compiled and preprocessed for consistency in spatial resolution, classification scheme, and temporal coverage. These datasets were subsequently analyzed via time series, enabling the discovery of trends and patterns from which future projections were calculated. Scenario modelling with Land Change Modeler (LCM) simulated potential land cover change under future scenarios, including “business-as-usual” and “ecological restoration” models. LCM integrates transition potential modeling, change prediction, and scenario analysis, and was calibrated using observed land cover transitions between 2010 and 2020. Random Forests machine learning algorithms were utilized to improve the predictability of future habitat changes and biodiversity gains and losses. Predictor variables included distance to water bodies, elevation, land cover history, and vegetation indices (e.g., NDVI). Feature importance analysis identified NDVI and proximity to open water as the most influential variables in predicting land cover transitions. Model performance was evaluated using several validation metrics: the overall classification accuracy was 89.2%, with a Kappa coefficient of 0.85 and an area under the ROC curve (AUC) of 0.91. K-fold cross-validation (k = 5) was also applied to assess model stability and generalization performance across data subsets.

We conducted spatial analysis tools that helped examine the land cover dynamics where it increased, decreased, or remained intact over time, thus fully grasping the potential ecological transitions. These were based on data in a predictive modelling framework that would protect the species and give insight into how we can conserve the species and monitor success/adaptive management strategies.

## 3. Results

### 3.1. Habitat Area Changes

The annual set of the LULCC maps covering the period between 2014 to 2023 depicts the temporal range of landcover types in the study area (shown in Figure 3). These maps, categorized into five primary habitats: Crops, Flooded Vegetation, Water, Range Land, and Bare Ground, exhibit the spatial distribution and extent of these classes over a decade.

Initially, the crop area appeared limited in 2014, with a noticeable expansion over the years, particularly from 2017 onward. By 2023, the crop areas show a significant increase, especially in the northern and central parts of the study region. This growth suggests intensified agricultural activities or conversion of other land types into agricultural land.

The flooded vegetation areas showed notable fluctuations. From 2014 to 2018, there is an increase in flooded vegetation. The water bodies (in blue) appear to be consistent in their extent but show some variability across the years. The period from 2016 to 2021 highlights a relatively stable distribution, with minor expansions in 2020 and 2021, possibly due to seasonal flooding or changes in water management practices. By 2023, the water areas show a slight reduction compared to earlier years.

Range land, identified in green, experienced dynamic changes, particularly during the mid-2010s. The area occupied by range land saw an initial increase from 2014 to 2017, followed by a gradual decrease towards 2023. The reported reduction in range land could be associated with an increase in agricultural activities  or with land management changes.

The barren land category seems quite stable, although it still shows minor annual variability. The most noticeable changes occur between 2014 and 2019, with a slight increase in some regions, possibly due to deforestation, land degradation, or the conversion of vegetated areas into non-productive land. However, by 2023, there is a minor reduction in bare ground, which could indicate efforts toward land restoration or natural vegetation regrowth.

### 3.2. Changes in Waterbird Metrics in Different Habitats

A total of 48 waterbird species were documented in the research area and categorized into 11 orders, 20 families, and 38 genera. The predominant taxonomic group observed in the research area was Charadriiformes, which consists of 13 species and represents 28.89% of the total waterbird species. Pelecaniformes and Anseriformes comprised 12 and 7 species, respectively, representing 26.67% and 15.56% of the total waterbird species. The Gruiformes order consisted of four species, accounting for 8.89% of the total species. All of the other six orders had proportions below 5%. The most prevalent species among the reported waterbirds were Anas acuta (Linnaeus, 1758) and Pulvialis fulva (Gmelin, 1789). There were 10 prevalent species, each with more than 500 individuals. These species include Anas acuta (Linnaeus, 1758), Pulvialis fulva (Gmelin, 1789), Porphyrio porphyrio (Linnaeus, 1766), Bubulcus ibis (Linnaeus, 1758), Dendrocygna javanica (Javan, 1819), Anas querquedula (Linnaeus, 1758), Anastomus oscitans (Boddaert, 1783), Himantopus himantopus (Linnaeus, 1758), Mycteria leucocephala (Latham, 1790), and Vanellus cinereus (Cretzschmar, 1827). The majority of these species belonged to the Anatidae and Charadriidae families.

The examination of the waterbird population and diversity in various environments between 2014 and 2023 provides valuable information about their environmental preferences and patterns in Figure 4a. The first graphic illustrates the fluctuating habitat preferences of waterbirds over the years. The flooded vegetation environment consistently had the highest proportion of waterbird abundance, followed by water, flooded vegetation, range land, crops, and bare ground. The waterbird population in the flooded vegetation environment experienced a minor decline from 2014 to 2017 but has remained mostly steady since 2018. The proportions of range land and water habitats remained largely steady, with a minor decrease in range land and a slight increase in water. Throughout the period, crops and barren land exhibited slight variations but consistently remained the least favored habitats. As illustrated in Figure 4a, the species richness data exhibited a comparable trend to abundance. The flooded vegetation environments had the greatest species richness, underscoring their significance for supporting various populations of waterbirds in Figure 4b. The range of land and water ecosystems also provided substantial biodiversity, but to a lesser degree. Crops and bare ground consistently represented the habitats with the lowest levels of diversity. Significantly, the number of species in flooded vegetation reached its highest point during 2017–2018, followed by a minor decrease and subsequent stabilization, which mirrored the trend in abundance.

The Shannon-Wiener diversity indices for waterbirds in various environments between 2014 and 2023 demonstrate substantial changes and discernible patterns in Figure 5a. In 2020, the aquatic habitat had its highest level of diversity, with a peak value of H′ = 4.00. Conversely, the lowest level of diversity was observed in 2018, approximately at H′ ≈ 1.07H’. Similarly, the diversity levels of the crops’ habitat exhibited variability, reaching its lowest point in 2018 (H′ ≈ 1.07) and peaking in 2020 (H′ = 2.14). The diversity of flooded vegetation experienced a downward trajectory over the years, reaching its peak in 2020 with a value of 2.85 (H′) and attaining its lowest point in 2018 with an approximate value of 1.00 (H′). The range of land exhibited variations, reaching its peak variety in 2020 (H′ = 2.99) and its lowest point in 2018 (H′ ≈ 1.06). The diversity of bare ground habitat exhibited significant variation, with the highest value in 2020 (H′ = 1.52) and the lowest value recorded in 2014 (H′ = 0.009).

Concerning the D_G–F_ indices, which serve as indicators of ecological conditions, comparable patterns of variation were detected across the different habitats in Figure 5b. The aquatic ecosystem, for example, exhibited a general downward trend, reaching its highest D_G–F_ value in 2020. On the other hand, the crop habitat showed varying D_G–F_ indices, reaching a minimum in 2014 and a maximum in 2021. The flooded vegetation displayed a declining pattern, reaching its peak D_G–F_ index in 2014 and its lowest point in 2023. The range land exhibited significant variations, with a decline until 2018 and a recovery by 2023. The bare ground habitat exhibited fluctuation, with its maximum D_G-F_ value occurring in 2020 and its lowest in 2014.

### 3.3. Waterbird Species and Population Changes in Water Metrics

The population and diversity of waterbirds exhibit fluctuations between 2014 and 2023. The population of waterbirds experiences fluctuations, with notable peaks occurring in 2015 and 2020. Similarly, the number of species present shows a comparable pattern, with distinct high points observed in 2015 and 2020. The Ardeidae family, which includes herons and egrets, had a significant presence with nine species, accounting for 20% of the overall species list. The Anatidae family exhibited the second highest level of diversity, consisting of seven distinct species, accounting for 15.56% of the total. Significant populations of notable species include Porphyrio porphyrio, Anas acuta, and Pulvialis fulva. Figure 6a displays data indicating a link between the abundance of waterbirds and the number of different species present in the Moe Yun Gyi Wetland throughout time. Figure 6b displays notable fluctuations in water diversion and water body area between 2014 and 2023. Figure 6c illustrates the correlation between water diversion, water body area, and water bird abundance. The regression study reveals a significant positive association between water body area and water diversion, with an R^2^ value of 0.4764. This indicates that higher levels of water diversion led to more significant water body areas. Waterbird abundance has a modest inverse relationship with water diversion, as indicated by an R^2^ value of 0.1013. This implies that increased water diversion may marginally impact waterbird abundance. Figure 6d illustrates the relationship between waterbird abundance, species richness, and water level. The regression study demonstrates an inverse correlation between the abundance of waterbirds and the water level, with an R^2^ value of 0.5676. This suggests an inverse relationship between water level and bird abundance, meaning that the number of birds tends to decline as the water level rises. There is a low positive connection between species richness and water level, as indicated by an R^2^ value of 0.0128. This suggests that water level has a minor impact on species richness.

An analysis was conducted to examine the relationships between changes in water levels and diversity indices for several types of waterbirds (see Figure 7). Dabbling birds and small waders displayed a marked negative relationship with higher water levels based on reduced Shannon-Wiener indices. These effects were even more pronounced at water levels close to 2.8 m, where the Shannon-Wiener indices averaged approximately 1.45. On the other hand, diving birds and large waders displayed a direct positive trend as water levels increased, with Shannon-Wiener indices peaking at around 2.5 m. In summary,  these results illustrate a positive relationship between water levels and the diversity and abundance of diving birds and large waders in the study area. Also, water levels maintained at approximately 1.5 m appear to facilitate a high Shannon-Wiener diversity index across all bird species, conducive to a favorable habitat for a diverse waterbird assemblage.

### 3.4. Changes in Rare Waterbird Species and Numbers in the Region

The rare waterbird species observed in the study area include Little Grebe (Tachybaptus ruficollis) (Podicipedidae) (Linnaeus, 1758), Cinnamon Bittern (Ixobrychus cinnamomeus) (Ardeidae) (Latham, 1790), Common Teal (Anas crecca) (Anatidae) (Linnaeus, 1758), Common Pintail (Anas acuta) (Anatidae) (Linnaeus, 1758), Ruddy Shelduck (Tadorna ferruginea) (Anatidae) (Pallas, 1764), Spot-billed Pelican (Pelecanus philippensis) (Pelecanidae) (Gmelin, 1789), Darter (Anhinga melanogaster) (Anhingidae) (Pennant, 1769), Common Duck (Sarkidiornis melanotos) (Anatidae) (Pennant, 1769), Cotton Pygmy Goose (Nettapus coromandelianus) (Anatidae) (Gmelin, 1789), Lesser Whistling Duck (Dendrocygna javanica) (Anatidae) (Latham, 1790), Spot-billed Duck (Anas poecilorhyncha) (Anatidae) (Forster, 1781), and Garganey (Anas querquedula) (Anatidae) (Linnaeus, 1758), and all are listed as Protected Area (PA). Additionally, the Philippine Duck (Pelecanus philippinus) (Pelecanidae) (Linnaeus, 1766) is classified as Near Threatened (NT) (IUCN, 2021), and the Great Crested Grebe (Podiceps cristatus) (Podicipedidae) (Linnaeus, 1758) is Vulnerable (VU) (IUCN, 2021). These species represent important conservation targets due to their status and ecological significance.

In total, 6 of the 48 waterbird species recorded in the study area are globally threatened according to the IUCN Red List. Of these, two species are Endangered (EN) (Calidris tenuirostris, Platalea minor), two are Vulnerable (VU) (Podiceps cristatus, Grus vipio), and one is Near Threatened (NT) (Limosa limosa). No critically endangered (CR) species were observed in the survey. These seven species comprise 15.6% of all species recorded. This number is significantly greater than the overall percentage of waterbird species at risk in Myanmar, which is 10.6%. More precisely, there is a single species of waterbird classified as endangered (EN), three species classified as vulnerable (VU), and three species classified as near threatened (NT). Using regression models, this study examined the correlation between the IUCN Red List of Threatened Species and time (Figure 7). The linear regression analysis explained the impact of independent variables (x) on the dependent variable (y), revealing trends in the protected bird population over time. Waterbirds classified as endangered (EN) exhibited notable variations. The numerical value began at 678 in 2014, reached its highest point at 872 in 2015, and had a significant decline to 44 in 2016. After recuperation and steadiness, the count experienced a substantial decline to 32 in 2023. Waterbirds classified as vulnerable (VU) also exhibited substantial fluctuations. The count commenced at 333 in 2014, experienced a marginal increase to 344 in 2015, and subsequently declined to 79 in 2016. After experiencing moderate volatility, the value eventually decreased to 74 in 2023. Waterbirds classified as Near Threatened (NT) have the smallest population size compared to other classifications. Commencing 29 in 2014, the tally declined to 9 in 2016 and exhibited slight variations, ultimately reaching 8 in 2023. The regression analysis indicated that the population of severely endangered waterbirds did not experience significant changes; however, the number of vulnerable and near-threatened waterbirds generally showed consistent growth over the years. The population of endangered waterbirds exhibited an increasing pattern. The category of vulnerable waterbirds experienced the most significant and striking modifications. These trends demonstrate the ever-changing nature of species classifications and the influence of conservation activities over the study period in Figure 8.

### 3.5. Dynamic Changes in Habitats, Waterbird Abundances, and Diversity Indices from 2030 to 2040

Between 2030 and 2040, projected habitat shifts show the land cover composition in the Moe Yun Gyi Wetland continuing to change as a result of an evolving ecological transition. Such environmental shifting is proposed to be influenced by agricultural expansion, variation in hydrological regime as a result of climate change, and more sedimentation or drainage with impacts on wetland hydrology. Consequently, stories of submerged vegetation and open water are expected to decrease, and reports of cropland and bare ground are expected to increase, especially in peripheral and northern areas of the wetland. These changes might have a great impact on the value and accessibility of the habitats for waterbirds and other bird species dependent on wetlands. We present these tendencies in Figure 9a, which shows a decrease in the bare ground along with an increase in flooded vegetation and water bodies. It shows a similar pattern to that of yet, hinting at a hydric state shift with more moisture holding capacity and vegetation cover from the early-2020s.

By 2040, this trend is anticipated to intensify, with wetlands becoming more composed of water and underwater plants. These changes suggest that previously degraded areas may be restored to a more natural wetland condition more favorable for a range of organisms. These changes may have far-reaching effects on local communities, playing a key role in fostering resilience in ecosystems and recovery of biodiversity.

The potential for both a decrease in bare ground and an increase in wetland habitats will likely lead to an increase in species richness. These changing landscapes will also differentially impact waterbird diversity as some species may take advantage of new water sources, while others will experience local shifts in habitat.

Outputs of modelling suggest modelling suggests that waterbird populations will stabilize by 2030, consistent with newly established wetland conditions. After this equilibrium phase, a synchronous but slow and steady increase in species richness and abundance is predicted to continue until 2040. This suggests a positive long-term ecological trajectory, with waterbird communities slowly latching on and becoming better suited to the more water-centric overall ecosystem.

Figure 9c shows Diversity Gain and Frequency (D_G–F_) and Shannon diversity index trends. D_G-F_ overall indices show considerable variability across both 2010 and 2020, particularly in bare ground and cropland habitats, indicating a phase of ecological disturbance and transitional states. Diversity responded to these fluctuations, but the overall Shannon diversity index was relatively stable, reflecting the persistence of biodiversity in modified, though still-natural, habitats. 

The D_G–F_ indices showed a declining trend between 2020 and 2030, followed by stabilization, indicating that the ecosystem went through a consolidation phase, resulting in relatively balanced and less varying environment. This stabilization allows wetland species to adapt to changes in habitat structure and is essential to ensure other wetland species cope with changes in their environment.

D_G–F_ and Shannon diversity indices will remain stable beyond 2040, whereas fluctuations in D_G–F_ values will remain high. Shannon diversity index values are projected to increase gradually after 2040. This indicates a gradual increase in habitat heterogeneity and species richness, which is conducive to a diverse and resilient wetland ecosystem.

Ecological outlooks for 2030 and 2040 highlight the importance of adapting conservation approaches to changing habitat conditions. Four recommended aspects of management intervention include sustaining habitat diversity, maintaining hydrological stability, and tracking biodiversity greyness and “specialness” in response to climate change. Such adaptive measures will be essential for the long-term sustainability of the Moe Yun Gyi Wetland as a conservation site and the ongoing protection of its ecological integrity in the East Asian–Australian Flyway.

## 4. Discussion

### 4.1. Ecological Conditions and Habitat-Specific Factors

The highest waterbird diversity was in 2020 when conditions for the waterbirds were most favorable. This could be due to favorable water quantities, good food availability, and minimum disturbances [11]. Maximum waterbird diversity was observed for the year 2020, when environmental conditions were optimal for a variety of ecological niches. This may be due to high water volume, food abundance, and low human interference. In particular, shallow to moderately deep water bodies at that time would have offered optimal foraging habitats for dabbling ducks, such as the Common Teal (Anas crecca [Linnaeus, 1758]) and the Northern Pintail (Anas acuta [Linnaeus, 1758]). Similarly, small waders such as Temminck’s Stint (Calidris temminckii [Leisler, 1812]) and Wood Sandpiper (Tringa glareola [Linnaeus, 1758]) used exposed mudflats and shallow waters. In contrast, diving birds, including grebes and cormorants, thrived in deeper open water areas that became more prominent during this year. Large waders, such as storks and herons, utilized the expanded wetland fringe zones, which likely supported abundant prey and experienced lower levels of human activity.

The diversity of flooded vegetation declined over the years and was highest in 2020. Flooded vegetation (vegetation that was flooded but not dead, such as a lake, river water, etc.) has also shown a rising trend (highest variation reached in 2020), but this category has decreased continuously in recent times. It is a habitat that is very sensitive to water fluctuation and hydrological conditions [9]. This indicates possible problems like water management practices, climate change impacts, or habitat loss. Rangeland showed considerable variation, with the highest diversity in 2020 and the lowest in 2018. These differences might also be associated with grazing pressures, changes in land use, or variability in vegetation cover that might affect waterbird diversity. Bare ground habitat had its peak diversity in 2020 and its lowest in 2014. Bare ground habitats tend to be less suitable habitats for waterbirds; they usually have fewer resources together with fewer vegetative covers that can modify habitat conditions. For this reason, fluctuations are probably associated with environmental conditions or anthropogenic interventions.

### 4.2. Waterbird Species, Abundances and Indices

Forty-eight species of waterbirds in eight taxonomic orders were fully recorded. Charadriiformes had the most representatives of all the orders. The variety in waterbirds, including the high densities of Anas acuta and Pulvialis fulva, reflect the importance of the wetland as a significant habitat for resident and migratory species. The existence of species such as *Anhinga melanogaster* and *Pelecanus philippinus* suggests the presence of expansive water bodies, vital for supporting a wide range of bird populations. The coexistence of both indigenous and migratory species, encompassing both reproductive and non-reproductive populations, suggests that the research site plays a crucial role as a habitat for aquatic birds all year round. The fluctuations in waterbird species and populations directly reflect the exploitation and quality of their habitat. Water level change in wetlands is a key factor influencing waterbird foraging and habitat selection [9,12,13]. Fluctuations in the water level of the marsh and flooded vegetation area will impact the abundance and diversity of waterbirds. From 2017 to 2020, there was a decrease in water diversion, followed by a change in hydrological conditions. The abundance of prominent and common species, such as *Anseriformes* and *Charadriiformes*, fluctuated significantly and consistently declined. The waterbird population has reached its highest level since the implementation of restoration efforts in 2017 due to the rise in water diversion. The fluctuations in the number of waterbird species between 2014 and 2023 primarily revolve around changes in the *Charadriiformes* group. This group is more responsive to variations in water levels and microhabitats compared to the *Anseriformes* group. The analysis of habitat preferences repeatedly showed that flooded vegetation had the most considerable abundance of waterbirds and species diversity. This environment’s significance is emphasized by its ability to support a wide variety and large number of waterbird populations, even in the face of minor changes in their abundance and diversity over time [14]. Rangeland and lake habitats provided more consistent conditions for waterbird populations, while crops and bare ground were the least favorable habitats. This highlights the fluctuation in numbers and species composition of waterbirds observed from 2014 to 2023, further illustrating the dynamic nature of the wetland ecosystem. The peaks in these parameters in 2015 and 2020 suggest that the ecological conditions promoting the increased stocks of waterbirds are repetitive. This shows a very high correlation between the size of the water body and water diversion, with an R^2^ value of 0.4764. This means that the more drainage there is, the more water body area there is. This suggests that more water diversion leads to an expansion of the water body area. Nevertheless, the modest inverse association between water diversion and waterbird abundance (R^2^ = 0.1013) indicates that excessive water diversion could potentially harm waterbird populations moderately. As observed for hard of the other major Asian wetlands, a positive trend in waterbird diversity has been observed at Moe Yun Gyi Wetland after rehabilitation [1]. For example, research conducted at Dongting Lake and Poyang Lake in China have found that species richness and bird abundances can be largely enhanced by water level management and wetland restoration activities [21,22]. Chilika Lake in India has seen a similar story of recovery of bird numbers and habitat use patterns, as a result of restoration actions such as hydrological reconnection and removal of an invasive species in recent years [23]. These results are consistent with the ecological results from Moe Yun Gyi, indicating the importance of an integrated restoration approach across Asian wetland ecosystems.

Between 2017 and 2020, reduced water diversion led to notable hydrological shifts, resulting in a consistent decline in the abundance of common species, particularly *Anseriformes* and *Charadriiformes*. These trends are similar to patterns observed in Dongting and Poyang Lakes in China, where waterbird populations declined in response to altered hydrological regimes [12,22]. Our findings also show that since restoration activities began in 2017, increased water diversion has corresponded with a recovery in waterbird populations, peaking in 2020—mirroring the success of wetland rehabilitation in Chilika Lake, India, where hydrological reconnection improved bird abundance [23]. The *Charadriiformes* group, more sensitive to water level fluctuations and microhabitat variability than *Anseriformes*, accounted for most of the interannual variation in species composition, aligning with findings from Mediterranean wetlands [14]. Our analysis consistently identified flooded vegetation as the habitat with the highest species richness and abundance, confirming previous research that emphasized its ecological value [9,13]. In contrast, rangeland and lake habitats offered more stable conditions, while crops and bare ground were the least favorable. The repeated peaks in waterbird abundance in 2015 and 2020 suggest that favorable hydrological conditions and habitat availability can cyclically restore bird populations, further supported by the moderate correlation between waterbody size and water diversion (R^2^ = 0.4764). However, the modest inverse relationship between waterbird abundance and water diversion (R^2^ = 0.1013) implies that excessive diversion, without concurrent habitat management, could dampen ecological gains.

According to this study, flooded vegetation and range land exhibited significant variability, indicating their susceptibility to alterations in the environment and management techniques. The Moe Yun Gyi Wetland serves as an important habitat for many species of waterbirds. The occurrence and behavior of some waterbirds can be tightly linked to the habitat’s condition and water management in this wetland. The quality of waterbird habitat and the diversity of their groups improve with the restoration of planted vegetation. This indicates a mix of resident and arrival species, with breeding and non-breeding populations covered by the data. This suggests that the research area is an essential habitat for a large number of waterbirds throughout the entire year. When the water diversion was limited to a volume of 1.35 × 10^6^ m^3^, the population of waterbirds were consistently stable. The overall ecological conditions reflected in the D_G–F_ indices closely mirrored the patterns observed in the Shannon-Wiener diversity indices. Overall, the water habitat showed a declining trend with a peak in 2020. The crops niche rose peaking in 2021, while flooded vegetation and range land showed downward trends. These patterns underscore the interdependence between habitat conditions and waterbird diversity, reflecting the roles of environmental variability and habitat-specific processes. Richness in dabbling birds and small waders showed a significant negative correlation with increasing water levels [9,16], thus suggesting lower diversity at high values around 2.8 m. In contrast, the negatively correlated groups, diver birds and big waders, presented a positive correlation; their richness was the maximum at 2.5 m. It shows that while diving birds and larger waders are better adapted to higher levels of water, dabbling birds and smaller waders thrive at lower water levels. Relative water levels kept at around 1.5 m support a great diversity of birds, which indicates strong conditions for the waterbird populations. This finding emphasizes the need for more flexible water-management strategies that foster the needs of different waterbird groups and promote ecological resilience. These results highlight the temporal fluidity of waterbird diversity and ecological parameters, driven by site-specific processes as well as broader environmental changes. These numbers highlight the need for ongoing monitoring and adaptive management for the conservation of biodiversity and continued resilience of waterbird populations under changing environmental conditions [9,17,18].

### 4.3. Fluctuations in the Population of Rare Species

Endangered (EN) waterbirds, including the Great Knot (*Calidris tenuirostris*) and Black-faced Spoonbill (*Platalea minor*)—two waterbirds depending on intertidal and shallow wetland habitats—showed significant decreasing trends during the study period. These declines are consistent with broader trends documented in the East Asian–Australasian Flyway, where species dependent on coastal wetlands have suffered due to habitat loss in key stopover sites, particularly in the Yellow Sea region [15,20]. Despite the restoration of the Moe Yun Gyi wetlands, a continued declining tendency for EN species population have been observed, indicating that local stressors are still influencing these populations.

Other species had stable or increasing trends, especially those that were habitat generalists. For example, populations of Common Teal (*Anas crecca*), Northern Pintail (*Anas acuta*), and Purple Swamphen (*Porphyrio porphyrio*) increased after 2018, possibly due to an increase in water availability and vegetation cover. This supports the findings of [9,17], who highlight the impact of shallow flooded vegetation on duck and rail diversity.

Species categorized as Vulnerable (VU) and Near Threatened (NT), such as the Great Crested Grebe (*Podiceps cristatus*), White-naped Crane (*Grus vipio*), Black-tailed Godwit (*Limosa limosa*), and Spot-billed Pelican (*Pelecanus philippensis*), showed either fluctuating or steadily declining trends. This demonstrates that even partial restoration of wetlands may still be inadequate to service the needs of specialized or migratory species, also seen in [19,24,25].

The distribution pattern well reflected the species level habitat preference. *Anas crecca*, *Anas poecilorhyncha*, and *Anas querquedula* preferred shallow waters with emergent plants, while waders such as *Tringa glareola*, *Pulvialis fulva*, and *Calidris temminckii* were more frequently appeared in exposed mudflats and seasonally flooded areas [24,25]. The preference for broader undisturbed water bodies by larger wading species like the Painted Stork (*Mycteria leucocephala*), Gray Heron (*Ardea cinerea*), and Great Egret (*Casmerodius albus*) was similar to the results of [12,26].

In conclusion, although habitat restoration at Moe Yun Gyi was beneficial in terms of rebound of certain widespread and eurytopic species, a more targeted conservation is needed in order to reverse the declines of the species at the risk of extinction (EN, VU, NT). This underscores the necessity of combining flyway-level protection with local wetland management to secure the long-term survival of threatened waterbird populations.

### 4.4. Discussion on Habitat Area Changes and Their Ecological Implications from 2010 to 2040

This study highlights the need for integrated land cover change projections and species distribution modeling to understand the ecological response to landscape change. The findings also illustrate the importance of adaptive conservation measures needed to maintain the presence of management-sensitive species in an ever-changing environment. Between the decades of 2010 and 2040, large extensive ecological phase transitions have already taken place, leading to various land cover changes. These changes have been the result of both natural processes, as well as anthropogenic actions, in efforts to rehabilitate and preserve wetland environments [25]. In the first decade (2010–2020), bare ground significantly expanded, most likely a result of deforestation, expansion of agriculture or other land-use changes. Flooded vegetation and rangeland areas, by contrast, increased, a trend that is consistent with hydrological changes related to flooding and land management practices that positively affect vegetative cover.

Big changes began to happen after 2020, when bare ground started to significantly decrease while the areas covered with water and vegetation increased. These findings indicate that intentional ecological restoration efforts and natural hydrological modulation have fueled wetland revitalization. These shifts reflect a transition to more aquatic-dominated ecosystems, facilitating better habitat quality and greater biodiversity. The first decade saw a distinct spread of bare ground areas. This may suggest deforestation, agricultural expansion, other land-use changes [18,27]. In contrast, flooded vegetation and range land areas increased, suggesting lessons from dynamic land-use types or natural biological processes. The prevalence of these vegetative habitats is indicative of hydrological changes, or land management practices, that favor vegetation [9]. In the next decade, the extent of the exposed ground took a dramatic decline while the area involving water and flooded vegetation increased. This suggests that extensive scenarios have been enacted to regenerate the ecology of the land or natural processes to allow once bare soil to convert into wetter and more vegetatively diverse settings. This regulatory transitional phase favoring more aquatic-dominated ecosystems may therefore represent a movement towards ecological making up for loss that both increase species habitat quality and biodiversity. By 2040, these trends will continue, areas of water and flooded vegetation will further expand, and areas of bare ground will continue to fall. The transition to water-dominated ecosystems would likely restructure local biodiversity, with implications for species composition and habitat availability. Waterbird abundance and species richness trends during the same period reflect the ecological impacts of these habitat changes. Population trends indicate high interannual variation with population booms from 2010 to 2020. Changes in habitat conditions, food availability, and climatic factors are probably reflected in these FLUXes. Richness remained fairly stable, which indicates that although the populations of the species fluctuated, the number of existing species was retained. This suggests a resilient aquatic bird community that can respond to changing conditions. The number of waterbirds significantly decreased from 2020 to 2030 and stabilized and slightly increased thereafter. This trend might suggest that early ecosystem response to habitat alteration occurs when waterbird populations adjust to new conditions. This initial reduction may be due to habitat disturbance or the time required for novel habitats to be suitable for supporting significant waterbird assemblages. In 2040, the ever-increasing diversity and abundance of waterbird species indicate that the waterbird populations will be recovering and adapting to the changed environment. This increase in numbers shows an increase in structure diversity and ecological quality, supporting a wider number of waterbird species. The ecological indices during these decades show the trend of changing habitat conditions and biodiversity. Here, D_G–F_ indexes, representing broader ecological conditions, showed the most notable variability over the first decade. These oscillations could refer to phases of ecological instability or transitional states due to land use changes and climatic variability. However, the shifts in these components were compensated by a relative stability in Shannon diversity indices during this period, indicating that the overall biodiversity of this region was sustained despite these changes, characteristic of a resilient ecosystem [24,26]. Between 2020 and 2030, the D_G–F_ indices showed a decrease, only to stabilize in continuity, which indicates a phase of ecological consolidation. This stabilization suggests a return to more typical ecological conditions, perhaps because of successful restoration work or simply a natural process of habitat development. Following stabilization, Shannon diversity indices exhibited slight improvements, reflecting the enhancement of habitat diversity and the overall number of species present. Also, both the D_G–F_ and Shannon diversity indices show similar patterns in the year 2040. The reduced variability of D_G–F_ indices suggests no major change in ecological stability. On the other hand, the continual rise in Shannon diversity indices indicates an overall improvement in biodiversity and resilience of the ecosystem to disturbances. This shows that the habitats can sustain diverse and robust biological communities, benefiting from ecosystem changes that increase vegetation and water dominance [16].

## 5. Conclusions

The restoration efforts at Myanmar’s Moe Yun Gyi Ramsar site have led to a remarkable sevenfold increase in waterbird populations, with species richness showing signs of stabilization. Particularly, the IUCN Red List has seen an increase in the number of uncommon birds, especially within the Pelecaniformes order, as a direct result of these conservation initiatives. The Shannon-Wiener diversity index peaked at 2.99 in 2020, indicating significant improvements in biodiversity. Enhanced marsh habitats, driven by strategic water management, including diversifying 25 million cubic meters of water, have been crucial in sustaining diverse and robust waterbird populations. Predictive models suggest that continued restoration efforts will further increase water and flooded vegetation areas while reducing barren ground by 2040, thereby improving habitat conditions for many species. However, these positive trends underscore the need for responsible water diversion management, guided by rainfall patterns and historical water data, to maintain and enhance habitat quality and biodiversity.

Future waterbird populations are expected to benefit from ongoing habitat improvements. However, continuous monitoring and adaptive management will be essential to ensure the long-term resilience of the Moe Yun Gyi wetland ecosystem. Ultimately, the Moe Yun Gyi Ramsar site stands as a testament to the significant impact that well-managed ecological restoration efforts can have on wetland conservation and the protection of waterbird species.

## Figures and Tables

**Figure 1 biology-14-00519-f001:**
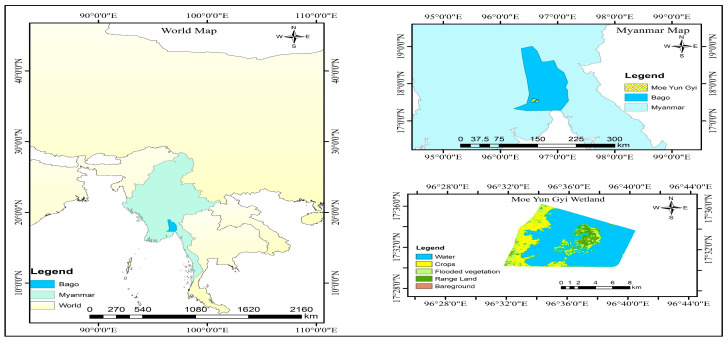
Study area map of Moe Yun Gyi constructed wetland.

**Figure 2 biology-14-00519-f002:**
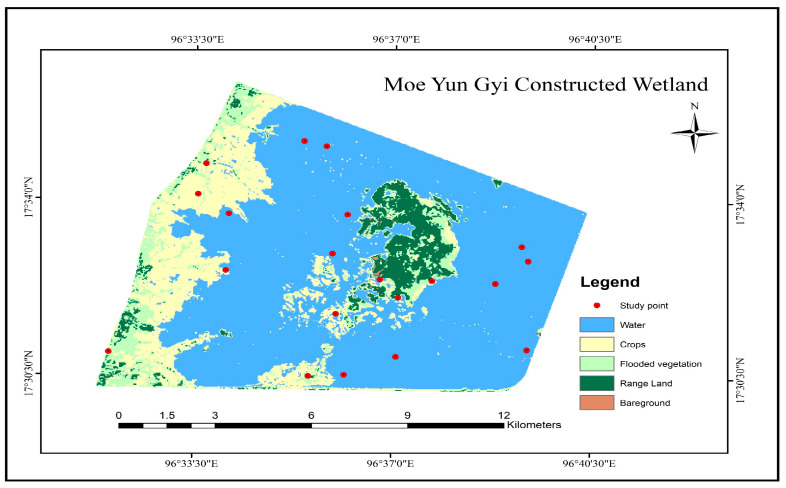
Study area with observation points.

**Figure 3 biology-14-00519-f003:**
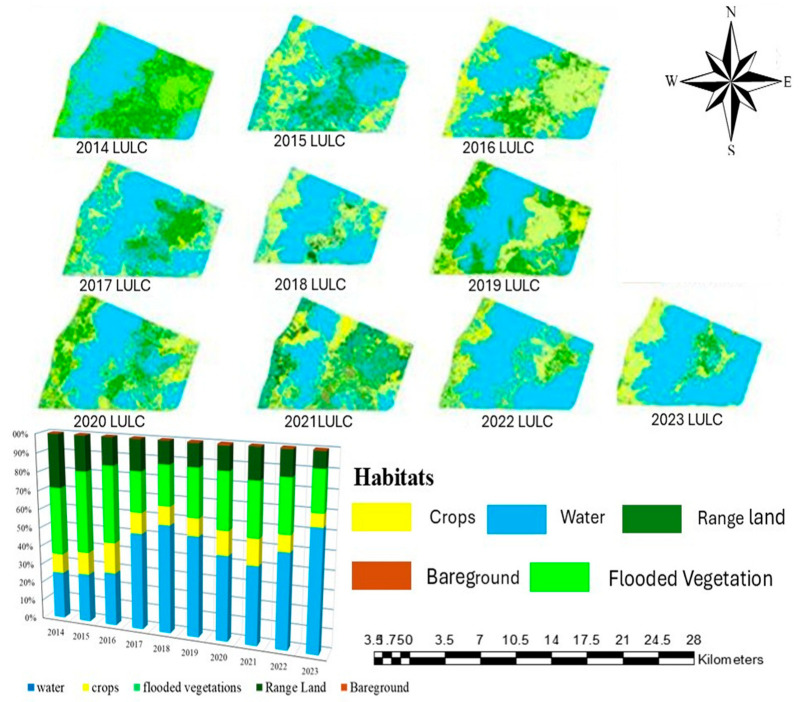
Surface area changes of habitats over years.

**Figure 4 biology-14-00519-f004:**
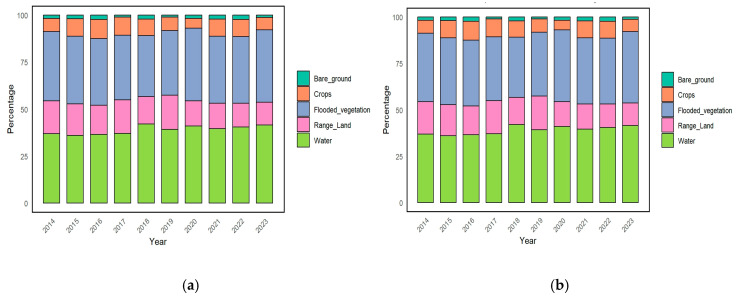
Proportion of species and numbers of waterbirds in different habitats over years. (**a**) Waterbird abundance percentage. (**b**) Waterbird species richness percentage.

**Figure 5 biology-14-00519-f005:**
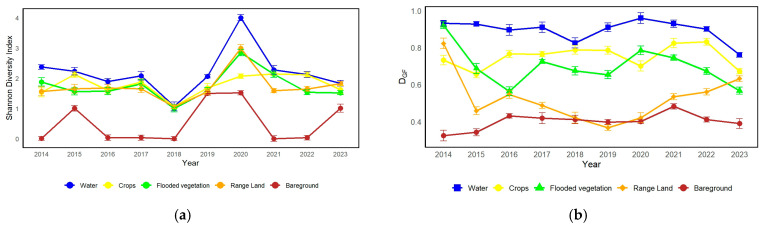
Annual variation of Shannon diversity indices and D_G-F_ indices of waterbirds in different habitats over years. (**a**) Shannon diversity indices. (**b**) D_GF_ indices.

**Figure 6 biology-14-00519-f006:**
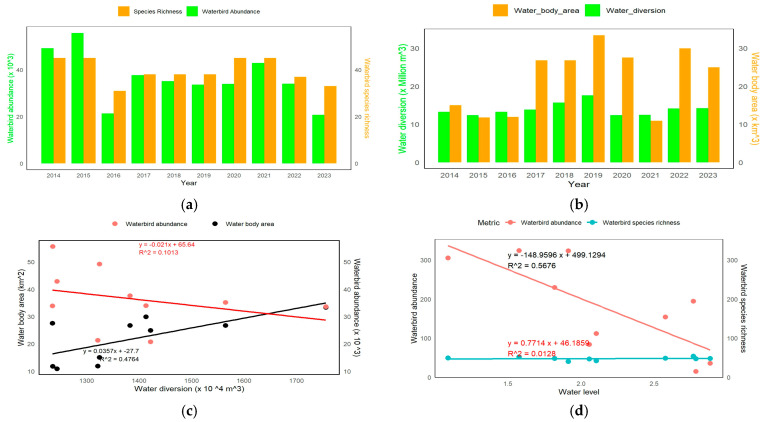
Annual dynamics of waterbird species richness and abundances in water metrics over years. (**a**) Waterbird abundance and species. (**b**) Water diversion and water body area. (**c**) Water diversion, water body area, and waterbird abundance. (**d**) Water level and waterbird metrics.

**Figure 7 biology-14-00519-f007:**
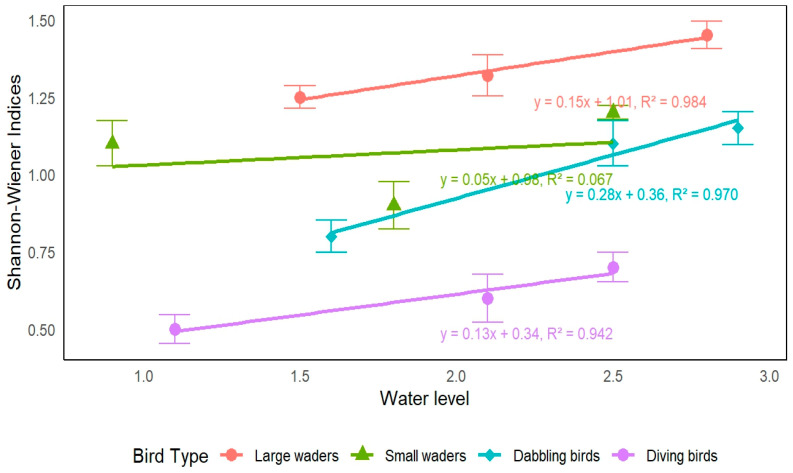
Effects of water level changes on Shannon-Wiener indices of four waterbird guilds.

**Figure 8 biology-14-00519-f008:**
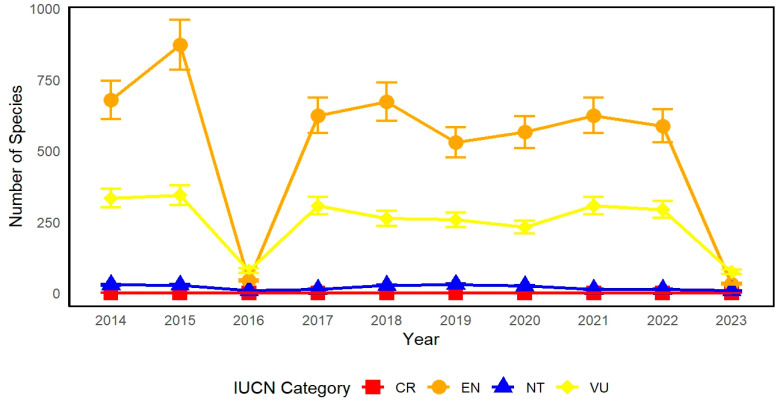
Bird numbers in relation to their IUCN red list status. Note: CR denotes critically endangered waterbirds; EN denotes endangered waterbirds; VU denotes vulnerable waterbirds; and NT denotes near threatened waterbirds.

**Figure 9 biology-14-00519-f009:**
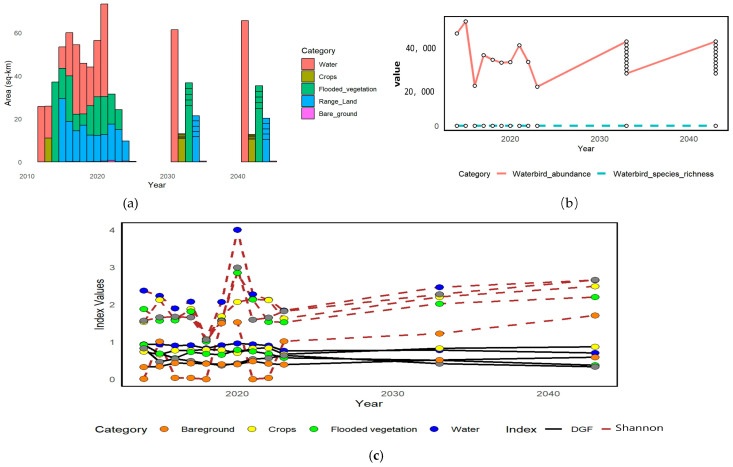
Prediction of dynamic changes in habitats, waterbird abundances and diversity indices over years. (**a**) Habitat area change. (**b**) Waterbird abundance and species richness. (**c**) D_G–F_ and Shannon Diversity Indices.

## Data Availability

The data that has been used is confidential.

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
