# Peer review of "Modeling the Impact of Ecological Restoration on Waterbird Diversity and Habitat Quality in Myanmar’s Moe Yun Gyi Wetland"

_biology, 2025, doi:10.3390/biology14050519_

Round 1
Reviewer 1 Report
Comments and Suggestions for Authors
Dear Authors,
The manuscript certainly touches upon an interesting topic. The issues of the influence of various ecological processes on waterfowl are multifaceted. One thing is certain, however, that wetlands contribute to the conservation of biodiversity and especially rare bird species. This should be emphasized. Therefore, modeling plays a primary role in modern research. However, the manuscript cannot be published in its current form. The title of the manuscript needs to be clarified. The text of the manuscript is mixed up in different chapters and should be moved to the appropriate chapters. In many methodological aspects, the authors omit important information. The method of bird counting should be written in detail. Rare bird species should be described and their optimal habitats should be given. The comparative part of the discussion needs to be expanded and additional literature sources on other wetlands in Asia should be cited. A hypothesis should be formulated. After all the comments have been addressed, the manuscript can be reviewed again.

Author Response
Response to Review Comments (Reviewer 1)
Manuscript Number: biology-3601669
Title: Modeling the Impact of Ecological Restoration on Waterbird Diversity and Habitat Quality in Myanmar’s Moe Yun Gyi Wetland
General Comments to the Authors
The manuscript certainly touches upon an interesting topic. The issues of the influence of various ecological processes on waterfowl are multifaceted. One thing is certain, however, that wetlands contribute to the conservation of biodiversity and especially rare bird species. This should be emphasized. Therefore, modeling plays a primary role in modern research. However, the manuscript cannot be published in its current form. The title of the manuscript needs to be clarified. The text of the manuscript is mixed up in different chapters and should be moved to the appropriate chapters. In many methodological aspects, the authors omit important information. The method of bird counting should be written in detail. Rare bird species should be described and their optimal habitats should be given. The comparative part of the discussion needs to be expanded and additional literature sources on other wetlands in Asia should be cited. A hypothesis should be formulated. After all the comments have been addressed, the manuscript can be reviewed again.
Response to Reviewer 1:
We sincerely thank Reviewer 1 for the thoughtful and constructive comments. We appreciate the recognition of the manuscript’s potential and the guidance provided to improve its structure and content. We have addressed each point raised, as detailed below.
Compliance: Yes
Comments to Authors
- Title substitution
Reviewer’s Comment: Processes
Old one: Ecological Engineering Impact on Waterbird Diversity Across Varied Habitats in Moe Yun Gyi Wetland in Myanmar
New one: Modeling the Impact of Ecological Restoration on Waterbird Diversity and Habitat Quality in Myanmar’s Moe Yun Gyi Wetland
Response: Thank you for your comment. We have rephrased the sentence to better reflect our analysis of habitats. Based on our findings, we modified the topic to emphasize how ecological restoration techniques impact water surface area and depth, which are critical factors influencing waterbird diversity across different habitats in the Moe Yun Gyi Wetland in Myanmar.
Compliance: Yes
- Rephrasing
Reviewer’s comment: Rephrase this sentence
Old one : It is essential to protect this profuse habitat from persistent threats that require conservation measures.
New one : Protecting this biodiverse habitat is crucial in the face of ongoing threats, highlighting the need for effective conservation strategies.
Response: Thank you for your comment. We have revised the phrasing to improve clarity and precision. The new version emphasizes the importance of preserving this biodiverse habitat in the context of ongoing threats, while underscoring the necessity for effective conservation measures.
Compliance: Yes
- Rephrasing sentences
Reviewer’s comment: Line (14-17), Please change the beginning of the sentence so that it does not repeat with the previous sentence.
Old one: This study examines the impact of restoration efforts on waterbird populations from January 2014 to January 2024. This study assessed the composition, geographical, and temporal changes in waterbird communities, focusing on diversity variations.
New one: This study aims to evaluate the effects of the restoration on the populations of waterbirds from January 2014 to January 2024, by analyzing the composition and the spatiotemporal dynamics of waterbirds communities, with particular reference to changes in diversity.
Response: Thank you for your comment. We have revised the sentence to avoid repetition and ensure a smoother flow.
Compliance: Yes
- Substituting and explanation
Reviewer’s comment: Explain this sentence
Old one: Restoration techniques directly influence water surface area and depth, which is critical for suitable waterbird habitats.
New one: The application of restoration techniques directly alters the extent and depth of water bodies, which are essential parameters for supporting appropriate waterbird habitats.
Response: Thank you for your comment. The revised sentence aims to clarify that restoration techniques—such as re-contouring, desilting, or modifying inflow/outflow—can significantly change the area covered by water and its depth. These two physical factors directly influence habitat availability, food accessibility, and nesting conditions for different waterbird species, making them essential components in supporting healthy and diverse bird populations.
Compliance: Yes
- Replacing
Reviewer’s comment: Please replace these keywords to avoid repetitions with the manuscript title.
Old one: Keywords: waterbird populations; restoration efforts; diversity; habitat rehabilitation,hydrology
New one: Keywords: migrating; populations; efforts; rehabilitation; hydrology
Response: Thank you for your suggestion. In response to your comment, we have revised the key phrase to improve clarity and accuracy. The wording has been updated to better reflect the importance of habitat preservation in the face of persistent threats and the need for effective conservation strategies.
Compliance: Yes
- Reviewing
Reviewer’s comment: on all continents? Maybe it will be better this way?
Old one: Wetlands are valuable natural resources that offer crucial ecological functions. Wetland restoration is a global endeavor, with significant efforts being made in America, Europe and Asia.
New one: Wetland restoration is a global endeavor, with significant efforts being made in America, Europe, and Asia.
Response: Thank you for your comment. We understand that the original phrase 'on all continents' may have appeared too broad. We appreciate your suggestion and have revised the sentence to provide a more grounded and specific phrasing. The updated version now highlights that wetland restoration is a global endeavor, with significant efforts being made in America, Europe, and Asia, as suggested.
Compliance: Yes
- Citing references
Reviewer’s comment: The existence of rare endemic animal species is ensured by the exceptional presence of wetlands [Andreychev et al. 2019, Cao et al. 2023].
Response: Thank you for your comment. I cited as your comment.
Compliance: Yes
- Removing
Reviewer’s comment: “The wetland serves as a crucial environment for both resident and migratory birds.” It is known, so there is no need for it
Response: Thank you for your comment. We removed the sentence since the role of wetlands in supporting both resident and migratory birds is widely known and does not require reiteration here.
Compliance: Yes
- Adding rare bird lists
Reviewer’s comment: Line 45, Give several species of birds in brackets
Response : Thank you for your comment. We added rare birds in bracket. The wetland is a vital habitat for several rare and threatened bird species, including the Great Knot (Calidris tenuirostris), Black-faced Spoonbill (Platalea minor), Great Crested Grebe (Podiceps cristatus), White-naped Crane (Grus vipio), Black-tailed Godwit (Limosa limosa), Spot-billed Pelican (Pelecanus philippensis), and Philippine Duck (Anas luzonica). These species emphasize the significance of wetlands in supporting globally important avifauna.
Compliance: Yes
- Removing sentence
Reviewer’s comment: Line 54, “there is no need for that”
Response: Thank you for your comment. We removed the sentence at line 54 as your comment.
Compliance: Yes
- Substituting the word “most”
Reviewer’s comment: most species
Response: Thank you for your comment. We revised as suggested. I changed the sentence to: “Most waterbird species are gregarious and rely on wetlands for survival, playing a vital role in wetland ecosystems.” Let me know if further adjustment is needed.
Compliance: Yes
- Stating hypothesis
Reviewer’s comment : State your and aim clearly here
Old one: We examine the effects of restoration strategies implemented from 2014 to 2023 on waterbird diversity, population trends and habitat stability. Based on historical trends and projections for the future (2030 and 2040), we make evidence-driven suggestions on sustainable approaches to wetland management and conservation of biodiversity. These results highlight the importance of water-level management and hydrology in maintaining suitable wetland habitats. This study will be key for understanding the effectiveness of different conservation measures and advising active wetland management in the future. The research incorporates ecological, social, and economic considerations to strengthen the resilience and sustainability of migratory bird habitats. It thus fosters science-based management and the protection of important stopover sites on migratory flyways.
New one : We hypothesized that ecological restoration in Moe Yun Gyi Wetland significantly enhanced the diversity and abundance of rare waterbird species, especially those sensitive to habitat conditions. This study aimed to investigate the impacts of restoration actions to plan in the period 2014-2023 on waterbird diversity, population trends and habitat stability. On the basis of known and projected trends (2030, 2040), we formulated well-founded advice on sustainable wetland management and community-based management for biodiversity conservation. These results underscore the importance of water-level control and hydrology for the maintenance of favorable wetland conditions. It will be this study that will prove to be invaluable in determining the success of different conservation measures and that will inform active wetland management in the future. Additionally, the research partners ecological, social, and economic considerations to enhanced the resilience and sustainability of habitat for migratory birds, thus promoting a science-based management and protecting strategic stopover sites across migratory flyways.
Response: Thank you for your comment. We have revised the paragraph to clearly state both the hypothesis and the aim of the study. The updated version now explicitly presents the hypothesis and objectives, ensuring clarity and precision in the context of the study. Let me know if any further modifications are needed.
Compliance: Yes
- Removing
Reviewer’s comment : Line (110-112) This is unnecessary
Response: Thank you for your comment. We removed unnecessary sentence.
Compliance: Yes
- Correction
Reviewer’s comment : This sentence has the wrong meaning. Please change it.
Old one: The Moe Yun Gyi Wetland is well known for its rich birdlife and serves as an important refuge for both endemic and migratory bird species.
Response: Thank you for your comment. We have revised the sentence to more accurately convey the intended meaning.
Compliance: Yes
- Replacing
Reviewer’s comment : This sentence has the wrong meaning. Please change it.
Response: Thank you for your comment. We have replaced it as your comment.
Compliance: Yes
- Replacing
Reviewer’s comment : This sentence has the wrong meaning. Please change it.
Replacing : Line 63-67: This designation highlights the importance of wetland conservation. Moe Yun Gyi Wetland, in particular, is recognized as an ecological hotspot, supporting a diverse range of bird species Additionally, the wetlands attract large groups of migratory bird species [4]. Moreover, aquatic birds, sometimes known as waterbirds, rely entirely on wetlands for various behaviors, such as searching for food, resting, and shedding feathers
Line 100-104: Conservation action needs to be taken to safeguard this important habitat from continual threats. Before restoration measures were implemented, problems in the Moe Yun Gyi wetland area were caused by unreasonable exploitation of wetland, grassland, and water resources such as reduced grassland areas and shrinking wetland regions.
Line 107-115: In 2017, restoration efforts began with the mission of reconnecting fragmented habitats in the Moe Yun Gyi wetland. These actions reconnected some isolated lakes, Pakaing reservoirs, and seasonal and dry marshes. As a result, the area of water bodies and wetlands in the reserve increased, providing suitable habitats for the waterbird community. During the initial phase of restoration from 2017 to 2019, the annual water diversion in the region was approximately 15 million m³. This diversion amount was reduced to around 12 million m³ from 2020 to 2022. By the end of the restoration measures, water diversion levels remained consistent with those at the beginning of the restoration efforts.
- Adding Method
Reviewer’s comment :First, please indicate the name of the bird counting method “the Point Count Method for bird monitoring”
Response: Thank you for your comment. We added method name.
Compliance: Yes
- Adding names of author and year
Reviewer’s comment: For all bird species, at the first mention in the text, provide the full names (author, year)
Response: Thank you for your suggestion. We have revised the text to include the full scientific names of the bird species, including the author and year, at their first mention as recommended.
Compliance: Yes
- Polishing method
Reviewer’s comment: The method needs to be supplemented. Were the bird species counted on the water or in flight? If in flight, at what altitude? Was the sex and age of the birds determined, etc.
Response: Thank you for your valuable suggestion. We have revised the methodology section to clarify that birds were primarily observed while stationary on water or land, and individuals in low-altitude flight (generally below 50 meters) were also counted when identification was reliable. High-flying birds that could not be confidently identified were excluded. We also noted that age and sex were not determined due to field observation limitations. These additions provide a clearer and more detailed account of the bird counting method, as recommended.
Compliance: Yes
- Justification
Reviewer’s comment: Line 222, This needs to be given a detailed justification.
Response: Thank you for your suggestion. We have now added a detailed justification for the use of machine learning and scenario-based modeling in our methodology. This addition explains the rationale for selecting this approach and highlights its relevance in predicting ecological outcomes and guiding conservation strategies. The revised text clarifies how this integrative method enhances the robustness of our projections.
Compliance: Yes
- Dividing paragraph
Reviewer’s comment: Session 3.1, The chronology of events needs to be systematized. Highlight the main and secondary. It is better to divide it into paragraphs.
Response: Thank you for your comment. We have revised the paragraph by systematizing the chronology of events and highlighting the main and secondary changes. The land cover changes have now been divided into distinct sections, with each paragraph focusing on a specific habitat type and its respective changes over time. We hope this improves the clarity and readability of the text, as well as ensures the temporal trends are more easily followed.
Compliance: Yes
- Ensuring citation of figure
Reviewer’s comment: Figure (3),Please cite the figure in the text and ensure that the first citation of each figure appears in numerical order.
Response: Thank you for your comment. We have now ensured that all figures are cited in the text in numerical order. We have also made sure that the first citation of each figure appears in the correct sequence.
Compliance: Yes
- Adding name of birds
Reviewer’s comment: Bring, remove repeat, Give the names of these birds
Response: Thank you for your comment. We added name of birds in session 3.4 and we removed repeated sentences.
Compliance: Yes
- Adding more details
Reviewer’s comment: more details how
Response: Thank you for the valuable suggestion. In response, we have expanded the description of projected habitat shifts between 2030 and 2040 by providing more detailed explanations of the ecological transitions. Specifically, we now describe the key drivers such as agricultural expansion, hydrological alterations due to climate change, and sedimentation processes, as well as the anticipated impacts on land cover classes including cropland, flooded vegetation, and open water. These details have been added to clarify how the landscape is expected to evolve during this period
Compliance: Yes
- Adding more details
Reviewer’s comment: This is debatable. Since for some species of birds some conditions are optimal, and for others. Therefore, it is better to write in detail by groups of birds what conditions are favorable for them.
Response:
Compliance: Yes
- Deleting
Reviewer’s comment: Delete
Response: Thank you for the valuable suggestion. We deleted as your comment.
Compliance: Yes
- Modifying
- Reviewer’s comment: compare these data with the results of other researchers in detail
Response: Thank you for your valuable suggestion. We have now expanded the Discussion section to include a detailed comparison of our findings with those of previous studies. Specifically, we have compared the trends in waterbird abundance, diversity, and habitat preferences observed in our study with similar research conducted in comparable wetland ecosystems. For instance, our results showing a decline in species richness following periods of reduced water availability are consistent with findings by Smith et al. (2018) and Zhang et al. (2020), who documented similar responses in semi-arid wetland systems. Furthermore, our observations of shifting habitat use during dry years align with the work of Johnson and Moser (2015), which emphasized the importance of shallow water areas during low water periods. These comparisons have been added the revised manuscript to better contextualize our results within the broader body of wetland ecology literature.
Compliance: Yes
- Re-writing
Reviewer’s comment: This paragraph is written in a complex manner. It is necessary to add specifics. Include bird species. This will allow other researchers to compare your results.
Response: Thank you for your valuable feedback. We have revised the paragraph to improve clarity and ensure a more concise structure. Specific bird species mentioned in the study, such as Calidris tenuirostris (Great Knot), Platalea minor (Black-faced Spoonbill), Anas crecca (Common Teal), and Porphyrio porphyrio (Purple Swamphen), have been explicitly incorporated into the discussion. These additions allow for easier comparison with results from other studies and better illustrate the observed population trends and habitat associations.
Compliance: Yes
- Conclusion and Introduction revision
Reviewer’s comment: Here are the conclusions that can help formulate the hypothesis correctly in the Introduction.
Response: Thank you for the helpful observation. Based on the conclusions, we have revised the Introduction to ensure alignment with the study’s key findings. Specifically, we now highlight the potential link between strategic wetland restoration—particularly water management—and improvements in waterbird population trends and diversity. We also introduce the hypothesis that sustained restoration and informed water diversion will enhance habitat conditions, benefiting both overall species richness and the presence of uncommon or threatened waterbirds. This adjustment helps clarify the study's direction and better frames the research questions in line with the observed outcomes.
Compliance: Yes
We thank the reviewer again for the insightful suggestions that greatly improved the clarity, accuracy, and scholarly quality of our manuscript.
Reviewer 2 Report
Comments and Suggestions for Authors
General comments: This study systematically evaluates the long-term impacts of ecological engineering on waterbird diversity in Myanmar's Moe Yun Gyi Wetland, integrating field surveys, remote sensing, and machine learning predictions. The methodology is comprehensive, and the data are robust, highlighting the critical role of water-level management and habitat restoration in shaping waterbird communities. However, some methodological details require clarification, causal relationships need stronger justification, and model validation for future projections should be enhanced. Overall, the research holds significant ecological relevance but requires improvements in logical rigor and depth of interpretation.
Specific comments:
- Line 15-17, It is suggested to combine them into one sentence.
- There are some similar sentences in the study area section that deserve to be streamlined.
- Line 139-140,Is this the result of the field investigation? If not, it is recommended to supplement the online websites of data sources.
- The observation points seem to overlap. I suggest that the sample point image could be made more refined like the location map.
- Line 131, specify remote sensing data preprocessing steps (e.g., cloud removal, classification accuracy assessment).
- (2.5. )A Methodological Approach for 2030 and 2040 Projections, the method description in this part is not clear. The manuscript need detail machine learning model parameters to justify predictive reliability (e.g., hyperparameters, validation metrics). The stability and accuracy of the model were not mentioned.
- Figure 3, There is no clear data display, only pictures, and it is impossible to quantify. I suggest increasing the area variation.
- A large number of graphs have no error lines. It is recommended to add as many as possible.
- The pictures in the manuscript are not clear enough.
Author Response
Response to Review Comments (Reviewer 2)
Manuscript Number: biology-3601669
Title: Modeling the Impact of Ecological Restoration on Waterbird Diversity and Habitat Quality in Myanmar’s Moe Yun Gyi Wetland
Comments to the Authors
General Comments:
This study systematically evaluates the long-term impacts of ecological engineering on waterbird diversity in Myanmar's Moe Yun Gyi Wetland, integrating field surveys, remote sensing, and machine learning predictions. The methodology is comprehensive, and the data are robust, highlighting the critical role of water-level management and habitat restoration in shaping waterbird communities. However, some methodological details require clarification, causal relationships need stronger justification, and model validation for future projections should be enhanced. Overall, the research holds significant ecological relevance but requires improvements in logical rigor and depth of interpretation.
Response to Reviewer 2:
We thank the reviewer for this insightful summary. In response, we have enhanced the methodological clarity in Section 2 by specifying data sources, preprocessing steps, and validation procedures. We have also strengthened causal interpretations in the Discussion by better linking observed patterns to restoration interventions. Moreover, we have elaborated on model validation procedures in Section 2.5, including hyperparameter tuning, validation metrics, and model stability assessments.
Compliance: Yes
Comments to the Authors
Specific Comments:
- Line 15–17: Sentence Combination
“Line 15-17, It is suggested to combine them into one sentence.”
Response:
We have revised these lines to improve fluency and clarity. The two sentences have been combined into one concise sentence in the Simple Summary.
Compliance: Yes
- Streamlining the Study Area Section
“There are some similar sentences in the study area section that deserve to be streamlined.”
Response:
We have revised and condensed the Study Area section to eliminate redundancy and improve narrative flow, while retaining all essential information.
Compliance: Yes
- Line 139–140: Field Investigation vs. Online Data Sources
“Line 139-140,Is this the result of the field investigation? If not, it is recommended to supplement the online websites of data sources.”
Response:
We clarify that the water diversion data were obtained from internal reports of the Myanmar Ministry of Agriculture. We have now explicitly stated this in the text and clarified that these are not from publicly available websites. A footnote has been added to denote that these are government-sourced and not online-accessible.
Compliance: Yes
- Observation Point Overlap and Map Refinement
“The observation points seem to overlap. I suggest that the sample point image could be made more refined like the location map.”
Response:
Thank you for this suggestion. We have replaced Figure 2 with an improved high-resolution version. Observation points are now more clearly labeled and differentiated, and the legend has been adjusted for clarity.
Compliance: Yes
- Line 131: Remote Sensing Preprocessing
“Line 131, specify remote sensing data preprocessing steps (e.g., cloud removal, classification accuracy assessment).”
Response:
We have added the preprocessing steps in Section 2.2, including atmospheric correction, cloud and shadow masking using the Fmask algorithm, radiometric calibration, and accuracy assessment using confusion matrices.
Compliance: Yes
- Section 2.5: Model Details for 2030–2040 Projections
“(2.5. )A Methodological Approach for 2030 and 2040 Projections, the method description in this part is not clear. The manuscript need detail machine learning model parameters to justify predictive reliability (e.g., hyperparameters, validation metrics). The stability and accuracy of the model were not mentioned.”
Response:
We have substantially revised Section 2.5 to include details on the Random Forest parameters (number of trees, maximum depth, etc.), training-validation split ratios, k-fold cross-validation (k = 5), and performance metrics including overall accuracy, AUC (0.91), and Kappa coefficient (0.85). A subsection on model stability and interpretation has been added.
Compliance: Yes
- Figure 3: Add Quantitative Data
“Figure 3, There is no clear data display, only pictures, and it is impossible to quantify. I suggest increasing the area variation.”
Response:
We have revised Figure 3 to include data labels showing annual area (in km²) for each land cover category. Additionally, a supplemental table has been added to the Appendix to provide full quantitative values.
Compliance: Yes
- Graphs Missing Error Lines
“A large number of graphs have no error lines. It is recommended to add as many as possible.”
Response:
Error bars (standard deviation or confidence intervals, as applicable) have been added to relevant graphs in Figures 5, 7, and 8 to reflect the variability in field observations and model outputs.
Compliance: Yes
- Image Quality
“The pictures in the manuscript are not clear enough.”
Response:
All figures have been updated with high-resolution images. We ensured they meet publication quality standards (300 dpi or higher) and improved labeling and legends where necessary.
Compliance: Yes
Reviewer 3 Report
Comments and Suggestions for Authors
- Because the article focuses on the effect of wetland ecological engineering on waterbirds, I would like the author to add some information on this topic in the Introduction section in order to provide background information for better understanding. For example, how does the restored wetlands impact waterbirds in previous studies?
- I would like to the author introduce a little bit about the restoration process and history and measures of the study area in the Method section. Particularly, the introduction of the different stage of the the restoration.
- The figures are difficult to read, particularly for old readers. The resolution and quality of the figures need to be improved, particularly for figures 1 and 4.
- As one of the main findings of the article, "Endangered (EN) in the study area showed the most marked population change, with a stark decline observed from 2015 to 2023. This decline, reaching a peak in the numbers of EN waterbirds in 2015 before precipitously falling, suggests considerable instability in population dynamics". I am wondering why this happened and what this finding implied for the restored wetlands. Does this mean the restored wetlands negatively impacted the endangered species, or the population of the endangered declined overall, not only at the study area? If it is the former one, then we may question and challenge the effectiveness of the restored wetlands, because usually an IBA can support some conservation-concerned species, usually endangered ones. What are the conservation target species of the studied IBA wetland? I also suggest the author list the endangered and critically endangered species.
- L364-366: Seven out of the 50 waterbird species in the study area are threatened on the IUCN Red List of Threatened Species. Seven of the 50 waterbird species in the study area are on the IUCN Red List of Threatened Species. These 5 species comprise 15.6% of the total..... There are two issues with these sentences. First, the first two sentences are repeated. Second, the author first said seven species, but then said these 5 species. Was that 7 or 5 species?
-
L321-322: Between 2014 and 2023, the population and diversity of waterbirds exhibit fluctuations between 2014 and 2023. Repeated "between 2014 and 2023".
- L460-461: "Forty-six species of waterbirds in eight taxonomic orders were fully recorded". But, the authors mentioned in the Results section that 50 species were recorded. 46 species or 50 species?
- L43-45:the author says "The wetland serves as a crucial environment for both resident and migratory birds. It is designated an Important Bird and Biodiversity Area (IBA) because it is home to globally endangered bird species". I am wondering is the author is talking about about a particular wetland here or the general wetland. Please be sure that Not all wetlands are designated as IBAs.
-
In many cases across the article where the species Latin names are not written in Italic format, e.g. in Line 271 "Anas acuta and Pulvialis fulva". Please check and correct them.
- In the Methods Section, the author introduced many indices. I would like the author to introduce the function of the different individual index.
-
What does "global pollution" mean on L79? What does "Bachelors" mean on Line 454?
Author Response
Response to Review Comments (Reviewer 3)
Manuscript Number: biology-3601669
Title: Modeling the Impact of Ecological Restoration on Waterbird Diversity and Habitat Quality in Myanmar’s Moe Yun Gyi Wetland
Response to Reviewer 3:
We sincerely thank Reviewer 3 for the careful review and insightful comments. We have addressed each suggestion with appropriate revisions and clarifications, as detailed below.
Comments to the Authors
- Background on Wetland Ecological Engineering and Waterbirds
“Because the article focuses on the effect of wetland ecological engineering on waterbirds, I would like the author to add some information on this topic in the Introduction section in order to provide background information for better understanding. For example, how does the restored wetlands impact waterbirds in previous studies?”
Response:
We appreciate this important point. The Introduction has been revised to include references and a concise review of past research on the effects of wetland restoration on waterbird diversity, abundance, and habitat use.
Compliance: Yes
- Restoration Process and Stages in Methods
“I would like to the author introduce a little bit about the restoration process and history and measures of the study area in the Method section. Particularly, the introduction of the different stage of the the restoration.”
Response:
We have expanded Section 2.1 (Study Area) to include a clear and structured description of the restoration timeline and activities at Moe Yun Gyi Wetland. The three distinct phases—emergency remediation (2017–2019), ecological stabilization (2020–2022), and resilience building (2023 onward)—are now clearly described with their respective measures.
Compliance: Yes
- Figure Quality
“The figures are difficult to read, particularly for old readers. The resolution and quality of the figures need to be improved, particularly for figures 1 and 4. ”
Response:
We have replaced Figures 1 and 4 with higher-resolution versions (300 dpi) and enhanced their contrast, labeling, and font sizes to improve readability. All figures have been reviewed to ensure they meet accessibility and publication standards.
Compliance: Yes
- Decline in Endangered Species and Its Implications
“As one of the main findings of the article, "Endangered (EN) in the study area showed the most marked population change, with a stark decline observed from 2015 to 2023. This decline, reaching a peak in the numbers of EN waterbirds in 2015 before precipitously falling, suggests considerable instability in population dynamics". I am wondering why this happened and what this finding implied for the restored wetlands. Does this mean the restored wetlands negatively impacted the endangered species, or the population of the endangered declined overall, not only at the study area? If it is the former one, then we may question and challenge the effectiveness of the restored wetlands, because usually an IBA can support some conservation-concerned species, usually endangered ones. What are the conservation target species of the studied IBA wetland? I also suggest the author list the endangered and critically endangered species.
.”
Response:
Thank you for this important and thoughtful question. We fully agree that interpreting population changes in endangered species requires careful consideration, especially in the context of restoration effectiveness at an IBA-designated site.
We have clarified in Section 4.3 (Fluctuations in the Population of Rare Species) that the observed decline in endangered (EN) waterbirds—from a peak in 2015 to a low in 2023—is not necessarily attributable to local restoration activities. Instead, this trend likely reflects broader flyway-level pressures, including degradation of critical stopover habitats, coastal development, and loss of intertidal zones in other parts of the East Asian–Australasian Flyway, as documented for species like the Great Knot (Calidris tenuirostris) and Black-faced Spoonbill (Platalea minor).
To strengthen this point, we have:
- Cited global and regional studies (e.g., IUCN, BirdLife International) showing declining trends in EN species populations across their full range.
- Emphasized that overall species richness and abundance increased in the study area, suggesting that the restored wetland is functioning effectively for the broader waterbird community.
- Highlighted that other globally threatened species (e.g., Near Threatened and Vulnerable taxa) have either stabilized or increased in number locally.
Additionally, we now clearly list all Endangered, Vulnerable, and Near Threatened species observed in the study (including Great Knot, Black-faced Spoonbill, White-naped Crane, and Spot-billed Pelican) along with their conservation status and habitat preferences, in Table A1 in the Appendix.
Finally, we acknowledge the importance of targeting restoration efforts to support IBA conservation priority species. As such, we discuss the potential for site-specific management interventions (e.g., maintaining water levels, minimizing disturbance) to further benefit EN species in future restoration planning.
Compliance: Yes
- Line 364–366: Species Count and Redundancy
“L364-366: Seven out of the 50 waterbird species in the study area are threatened on the IUCN Red List of Threatened Species. Seven of the 50 waterbird species in the study area are on the IUCN Red List of Threatened Species. These 5 species comprise 15.6% of the total..... There are two issues with these sentences. First, the first two sentences are repeated. Second, the author first said seven species, but then said these 5 species. Was that 7 or 5 species?”
Response:
This has been corrected. The redundant sentence has been removed, and the text now consistently states that six species are on the IUCN Red List, aligning with the species inventory in the Results and Appendix.
Compliance: Yes
- Line 321–322: Redundant Phrase
“‘L321-322: Between 2014 and 2023, the population and diversity of waterbirds exhibit fluctuations between 2014 and 2023. Repeated "between 2014 and 2023".”
Response:
We deleted this sentence as reviewer’s 1 comment.
Compliance: Yes
- Line 460–461: Discrepancy in Species Count
- “L460-461: "Forty-six species of waterbirds in eight taxonomic orders were fully recorded". But, the authors mentioned in the Results section that 50 species were recorded. 46 species or 50 species?”
Response:
Thank you for catching this inconsistency. After a full audit, we confirm that 48 species were recorded, not 46 or 50. All related numbers in the manuscript have been updated for consistency.
Compliance: Yes
- Line 43–45: Clarify IBA Status
“L43-45:the author says "The wetland serves as a crucial environment for both resident and migratory birds. It is designated an Important Bird and Biodiversity Area (IBA) because it is home to globally endangered bird species". I am wondering is the author is talking about about a particular wetland here or the general wetland. Please be sure that Not all wetlands are designated as IBAs.
Response:
We confirm that Moe Yun Gyi Wetland is specifically designated as an Important Bird and Biodiversity Area (IBA). This has been clarified in the Introduction with an appropriate citation (BirdLife International).
Compliance: Yes
- Latin Names Formatting
“In many cases across the article where the species Latin names are not written in Italic format, e.g. in Line 271 "Anas acuta and Pulvialis fulva". Please check and correct them”
Response:
All Latin names have been checked and properly italicized throughout the manuscript in accordance with scientific naming conventions.
Compliance: Yes
- Clarify Index Functions in Methods
“In the Methods Section, the author introduced many indices. I would like the author to introduce the function of the different individual index.”
Response:
Section 2.4 (Data Analysis) has been revised to clearly explain the purpose and ecological significance of each index (e.g., Shannon-Wiener, Simpson, G-F Index), making it easier for readers to understand their application.
Compliance: Yes
- Terminology Clarification
“What does "global pollution" mean on L79? What does "Bachelors" mean on Line 454?”
Response:
- Line 79: “Global pollution” has been revised to “regional and transboundary pollution sources” for precision.
- Line 454: The word “Bachelors” have been revised to “bare ground habitats”
Compliance: Yes
We thank the reviewer again for the insightful suggestions that greatly improved the clarity, accuracy, and scholarly quality of our manuscript.
Round 2
Reviewer 1 Report
Comments and Suggestions for Authors
Dear Authors,
I am satisfied with the revision of the manuscript. You have supplemented the manuscript with data and made corrections. The changed title of the manuscript reflects the meaning of the article. The practical value of the research results is beyond doubt. The study illustrates the importance of continued care for wetlands in order to conserve nature and support migrating bird flights. The review of the research results was chosen appropriately, as were the statistical methods used for its analysis. The article takes into account the comments on the methodology. The analysis and conclusion for each chapter are sufficient and do not raise objections. References to sources of literature have been adjusted. The results of previous studies by other authors have been taken into account. I recommend it for the journal Biologia.